# Factor-Dependent Internal Ribosome Entry Site and -1 Programmed Frameshifting Signal in the Bemisia-Associated Dicistrovirus 2

**DOI:** 10.3390/v16050695

**Published:** 2024-04-28

**Authors:** Yihang Chen, Subash Chapagain, Jodi Chien, Higor Sette Pereira, Trushar R. Patel, Alice K. Inoue-Nagata, Eric Jan

**Affiliations:** 1Department of Biochemistry and Molecular Biology, Life Sciences Institute, University of British Columbia, Vancouver, BC V6T 1Z3, Canada; yihangchen1013@gmail.com (Y.C.); chsubash@student.ubc.ca (S.C.); chienjodi@gmail.com (J.C.); 2Alberta RNA Research and Training Institute, Department of Chemistry and Biochemistry, University of Lethbridge, Lethbridge, AB T1K 3M4, Canada; higor.pereira@uleth.ca (H.S.P.); trushar.patel@uleth.ca (T.R.P.); 3Embrapa Vegetables, Brasília 70770-901, Brazil; alice.nagata@embrapa.br

**Keywords:** virus, frameshifting, RNA, IRES

## Abstract

The dicistrovirus intergenic (IGR) IRES uses the most streamlined translation initiation mechanism: the IRES recruits ribosomes directly without using protein factors and initiates translation from a non-AUG codon. Several subtypes of dicistroviruses IRES have been identified; typically, the IRESs adopt two -to three overlapping pseudoknots with key stem-loop and unpaired regions that interact with specific domains of the ribosomal 40S and 60S subunits to direct translation. We previously predicted an atypical IGR IRES structure and a potential -1 programmed frameshift (-1 FS) signal within the genome of the whitefly Bemisia-associated dicistrovirus 2 (BaDV-2). Here, using bicistronic reporters, we demonstrate that the predicted BaDV-2 -1 FS signal can drive -1 frameshifting in vitro via a slippery sequence and a downstream stem-loop structure that would direct the translation of the viral RNA-dependent RNA polymerase. Moreover, the predicted BaDV-2 IGR can support IRES translation in vitro but does so through a mechanism that is not typical of known factorless dicistrovirus IGR IRES mechanisms. Using deletion and mutational analyses, the BaDV-2 IGR IRES is mapped within a 140-nucleotide element and initiates translation from an AUG codon. Moreover, the IRES does not bind directly to purified ribosomes and is sensitive to eIF2 and eIF4A inhibitors NSC1198983 and hippuristanol, respectively, indicating an IRES-mediated factor-dependent mechanism. Biophysical characterization suggests the BaDV-2 IGR IRES contains several stem-loops; however, mutational analysis suggests a model whereby the IRES is unstructured or adopts distinct conformations for translation initiation. In summary, we have provided evidence of the first -1 FS frameshifting signal and a novel factor-dependent IRES mechanism in this dicistrovirus family, thus highlighting the diversity of viral RNA-structure strategies to direct viral protein synthesis.

## 1. Introduction

RNA viruses carry genomes with limited coding capacity. As such, a key essential step in the viral replication cycle is commandeering the host translational machinery to direct viral protein synthesis. In eukaryotes, protein synthesis is initiated via the cap-dependent scanning model, whereby ~10 core translation factors mediate recruitment of the small 40S ribosomal subunit to the 5′cap and scanning [1]. RNA viruses have adapted unique strategies to recruit and manipulate the host ribosome, some of which utilize an internal ribosome entry site (IRES), which is a cis-acting RNA element, typically structured, that can mediate ribosome recruitment in a 5′-end independent manner and using a subset of translation initiation factors and/or other cellular proteins. Currently, viral IRESs can be grouped into six distinct classes based on the conservation of RNA structure and factor requirements; however, with the expanding viromes identified via metagenomic approaches [2,3], it is anticipated that many more classes have yet to be discovered.

The dicistrovirus intergenic IRES studied to date are the most streamlined translation mechanism to date [4]. Dicistrovirus contains a ~8–10 kb viral RNA genome encoding two main open reading frames, ORF1 and ORF2. ORF1 encodes the viral non-structural proteins such as replicase, protease, and RNA helicase, whereas ORF2 encodes the structural proteins. The ORFs are translated by distinct IRESs, the 5′UTR and the IGR IRES. The 5′UTR IRES utilizes a Type III-like IRES mechanism requiring several translation factors [5]. The model IGR IRES, exemplified by that in cricket paralysis virus (CrPV), adopts a triple-pseudoknot RNA structure that can directly recruit the 40S ribosomal subunit, and then the 60S, to form the 80S ribosome and initiate translation from a non-AUG codon. Some IGR IRESs can recruit preformed 80S to direct translation, including a fraction of CrPV IGR IRES [6,7,8]. Cryo-EM studies of IRES:ribosome complexes captured in the translocation cycles revealed that distinct domains of the IRES interact with specific parts of the ribosome in order to manipulate the ribosome into a primed elongation mode of translation [9,10,11,12,13]. Within the IGR IRESs, there are distinct subtypes with specific RNA structures; they adopt a similar RNA structure as the CrPV IGR IRES to direct translation [13,14]. However, it is evident that there may be other more atypical IGR IRES mechanisms within dicistroviridae. The IGR from Halastavi árva virus (HalV), isolated from the intestinal contents of freshwater carp (*Cyprinus carpio*) [15], was identified to contain novel IRES features [7]; the HaIV IGR IRES lacks key domains including PKIII, SLIV, and SLV. Moreover, the HaIV IGR IRES does not recruit the 40S subunit but instead binds to preformed 80S ribosomes to drive translation. The discovery of HaIV IGR IRES and other similar IRESs leads to the possibility of alternative IGR IRES mechanisms within the family *Dicistroviridae*. 

Virome metagenomics studies have identified novel dicistrovirus-like genomes, which provide an opportunity to address the repertoire of IRES mechanisms that drive protein synthesis in this viral family [2,3]. A recent report by Hedil et al. identified a novel dicistrovirus genome with unique features. Bemisia-associated dicistrovirus 2 (BaDV-2) was isolated from the sweet potato whitefly *Bemisia tabaci* [16]. The 8012 nt genomic sequence of BaDV-2 contains (Accession MN231041) three ORFs (ORF1a, ORF1b, and ORF2) with a 5′UTR, an intergenic region (IGR), a 3′UTR, and a poly(A) tail. ORF1a contains an RNA helicase (nt 1780-2121) and a 3C cysteine protease motif (nt 3541-3654). ORF1b is in -1 frame and contains an RdRp motif (nt 4047-5261). ORF2 contains two picornavirus capsid protein-like Rhv motifs (nt 5893 to 6177 and 6730–7002, respectively), a calicivirus coat-protein motif (nt 7021 to 7209), and a CrPV-capsid protein-like motif (nt 7336 to 7977). A putative -1 frameshifting signal (-1 FS) was proposed to drive the translation of the full-length ORF1 polyprotein, consisting of a slippery sequence (GUCUUUU, nt 3780-6) and a putative pseudoknot, whereas a putative IGR IRES was proposed to drive the translation of ORF2. 

In this study, we demonstrated that the BaDV-2 IGR supports IRES activity using a bicistronic reporter in vitro in insect *Spodoptera frugiperda* (Sf) 21 and rabbit reticulocyte lysate (RRL) translation extracts. Deletion and mutagenesis analyses mapped the BaDV-2 IRES activity within a 140-nucleotide element containing a predicted stem-loop. Moreover, the IRES is sensitive to eIF2 and eIF4A inhibitors NSC1198983 and hippuristanol, respectively, indicating that this IRES is factor-dependent. Finally, we showed that the BADV-2 putative -1 FS uses a stem-loop structure and slippery sequence to support 4% and 8% -1 frameshifting in Sf-21 and RRL, respectively. In summary, we have provided evidence of a novel IRES and -1 FS mechanism in the dicistrovirus family, thus highlighting the diversity of viral strategies to direct viral protein synthesis.

## 2. Materials and Methods

### 2.1. Cell Culture

*Drosophila* Schneider 2 cell line (S2) cells (Invitrogen) were maintained in Shields and Sang M3 insect medium (Sigma-Aldrich, St. Louis, MI, USA) supplemented with 10% fetal bovine serum at 25 °C.

### 2.2. Plasmids

BaDV-2 IGR IRES fragment (nucleotides 5325 to nucleotides 5694, Accession No. MN231041) (Twist Biosciences) was cloned into bicistronic and monocistronic luciferase reporter plasmids using Gibson assembly. Deletion mutations and specific nucleotide mutations were generated by PCR-based methods. BaDV-2 IGR IRES with a downstream P2A peptide sequence (Twist Biosciences) was cloned into the CrPV infectious clone [17], replacing the CrPV IGR IRES.

### 2.3. In Vitro Transcription

Monocistronic, bicistronic, and infectious clones were linearized by NarI, BamHI, and Eco53KI (NEB) restriction enzymes, respectively. RNA was transcribed using T7 RNA polymerase and subsequently purified with an RNeasy kit (Qiagen). 5′capping and polyadenylation were performed post-transcriptionally (CellScript). The RNA integrity and purity was confirmed by denaturing agarose gel analysis, and concentration was measured with a spectrophotometer (Nanodrop).

### 2.4. Transfection

S2 cells (3 × 10^6^ cells) were transfected with bicistronic or infectious clone RNA (3 μg) using Lipofectamine 2000 reagent (Thermo Fisher Scientific, Waltham, MA, USA). For monitoring reporter RNA translation under CrPV infection, S2 cells were transfected with bicistronic RNA for 1 h, followed by infection with CrPV (MOI = 20) for 4.5 h. For infectious clone transfection experiments, S2 cells were transfected and then harvested at the indicated times and lysed in 1× passive lysis buffer (Promega, Tokyo, Japan). Lysates were cleared and protein concentration was measured by Bradford assay (Bio-Rad, Hercules, CA, USA). Western blotting was performed using an anti-VP2 antibody.

### 2.5. In Vitro Translation Reactions

In vitro transcribed bicistronic (1 µg) and infectious clone (2 µg) RNAs were incubated in Sf-21 cell extract (Promega) or rabbit reticulocyte lysate (RRL) (Promega) for 2 h or 45 min, respectively, in the presence of [^35^S]-methionine/cysteine. Inhibitors (hippuristanol or NSC119889) at appropriate concentrations were added to the reaction 5 min prior to the addition of RNA. Reactions were either loaded and resolved on 15% SDS-PAGE gels and analyzed by phosphorimager analysis (Typhoon, Amersham, Chicago, IL, USA) or analyzed for enzymatic luciferase activity (Promega) using a luminometer (Turner Designs TD-20/20, San Jose, CA, USA).

### 2.6. Ribosome Filter Binding Assay

The 40S and 60S ribosomal subunits were purified as previously described [18]. RNA was dephosphorylated by FastAP thermosensitive alkaline phosphatase (Thermo Fisher Scientific) then labeled with T4 polynucleotide kinase (NEB) and [γ^32^P]-ATP. [γ^32^P]-RNA (final concentration: 0.5 nM) was heated at 65 °C for 3 min prior to the addition of 1× buffer E (final concentration: 20 mM Tris pH 7.5, 100 mM KCl, 2.5 mM MgOAc, 0.25 mM Spermidine and 2 mM DTT) and gently cooled to room temperature for 20 min, allowing for proper folding. The RNA was then incubated with purified 40S ribosomal subunits from 0.1 nM to 100 nM for 5 min, followed by 60S ribosomal subunits from 0.15 nM to 150 nM for 15 min, with 50 ng/mL in vitro transcribed noncompetitor RNA prepared from pcDNA3 (linearized with EcoRV) to prevent non-specific binding [19]. Reactions were loaded to the Bio-Dot filtration apparatus (Bio-Rad) with nitrocellulose and nylon membranes (Zeta-Probe, Bio-Rad). The membranes were then washed three times with 1× buffer E, then dried and imaged by phosphoimager analysis (Amershan Typhoon, ImageQuant™ 800). The fraction bound and the dissociation constant (K_D_) were calculated as previously described [18]. 

### 2.7. RNA Purification and Size Exclusion Chromatography with Multi-Angle Light Scattering (SEC-MALS)

In vitro transcribed BaDV-2 IGR IRES RNA was purified by size exclusion chromatography (SEC) with an ÄKTA pure fast protein liquid chromatography (FPLC) (Global Life Science Solutions, New York, NY, USA) as previously described [20]. RNA-containing fractions were then pooled and concentrated by ethanol precipitation with resuspension in HEPES folding buffer (final concentration: 50 mM HEPES, 150 mM NaCl, 15 mM MgCl2, 3% Glycerol, and at pH 7.4) for SEC-MALS analysis and small-angle X-ray scattering (SAXS) analysis at the concentration of 2 mg/mL. SEC-MALS analysis was performed as previously described with the Optilab Refractive Index System (Wyatt Technology) [20].

### 2.8. Small-Angle X-ray Scattering (SAXS) and Atomic Structure Calculation

SAXS was performed as previously described by D’Souza et al. [20]. Briefly, SAXS data for BaDV-2 “minimal” IRES RNA were collected at 2 mg/mL using HEPES folding buffer. Samples were run at Diamond Light Source Ltd. synchrotron (Didcot, UK) on the B21 SAXS beamline, with a high-performance liquid chromatography (HPLC) system attached upstream to ensure the monodispersity [21]. A specialized flow cell was employed in conjunction with an inline Agilent 1200 HPLC system (Agilent Technologies, Stockport, UK). Then, 50 μL of RNA samples were injected onto a Shodex KW403-4F (Showa Denko America Inc., New York, NY, USA) size exclusion column with a flow rate maintained at 0.160 mL/minute. The eluted samples were exposed to X-rays with 3 s exposure time and 600 frames. Analysis of scattering data was performed using the ATSAS suite as previously described [22,23]. Twenty RNA models were generated using DAMMIN, averaged and filtered to a single representative using DAMAVER.

We utilized MC-Fold [24] to anticipate multiple low-energy secondary structures for BaDV-2 IRES RNA. This was performed to produce secondary structure predictions for downstream use. In each case, we chose the structures with the lowest energy as input files for MC-Sym, an application that facilitates the reconstruction of three-dimensional structures using known structures based on fragments [24]. Using MC-Sym, we computed 100 all-atom structures for each RNA and minimized them using MC-Sym’s implemented protocols. These minimized structures were then compared to experimental SAXS data with CRYSOL based on minimized structures, to determine Rg and the goodness-of-fit parameter (χ2) [25]. To subject the minimized structures to CRYSOL, we used the protocols implemented in MC-Sym. Finally, we ranked the MC-Sym-derived structures with low-resolution structures based on their χ2 values and aligned them using the program SUPCOMB [26].

### 2.9. SHAPE Probing

For SHAPE-MaP analysis of the “minimal” IRES, in vitro transcribed RNA for BaDV-2 IRES was used as described [27]. Briefly, 500 ng of RNA was heated to 95° for 3 min, followed by the addition of Buffer E (final concentration of 20 mM Tris, pH 7.5, 0.1 M KCl, pH 7.0, 2.5 mM MgOAc, 0.25 mM spermidine and 2 mM dithiothreitol (DTT)) and incubated at 30 °C for 20 min. Folded RNA was modified by adding N-methylisatoic anhydride (NMIA) dissolved in DMSO (final concentration 5 mM) and incubated for 45 min at 30 °C. Control reactions containing only DMSO (no NMIA), as well as additional denaturing control (DC) reactions were performed in parallel. The RNA was recovered by ethanol precipitation.

RT Primer extension of modified and control RNAs was performed with primer 5′ -CCTTCACTTTTAACATGGTTGGCCTG-3′ using Superscript III Reverse Transcriptase (SSIII, Thermo Fisher Scientific) with Mn^2+^, followed by second-strand cDNA synthesis (NEB #E6111). The ds cDNA was then used for subsequent library preparation and sequencing. Libraries were sequenced on an Illumina MiSeq platform following the manufacturer’s standard cluster generation and sequencing protocols (UBC Sequencing and Bioinformatics Consortium).

SHAPE-Mapper 2.0 was used to analyze the sequencing data as described [28]. RNAstructure [29] was used to predict and model the secondary structure using the SHAPE-MaP data.

## 3. Results

### 3.1. BaDV-2 Contains a Programmed -1 Frameshift (-1 FS) Signal

A pseudoknot structure with an upstream putative slippery sequence “GUCUUUU”, which are characteristics of a -1 programmed frameshift signal (FS), was proposed within the BaDV-2 genome [16,30] (Figure 1A). To determine whether this element can support frameshifting, we cloned the putative -1 FS element (nucleotides 3742 to 3884, Accession MN231041) into a bicistronic reporter between the Renilla (RLuc, 0 frame) and Firefly luciferase (FLuc, -1 frame) open reading frames (Figure 1B). We cloned sequences upstream and downstream of the predicted -1 FS signal to ensure the entire element is present. We also generated an in-frame control reporter whereby the RLuc and FLuc ORFs are in the same frame and a mutation (UUU3783-85 to CCC) (SS MUT) that disrupts the putative slippery sequence from “GUCUUUU” to “GUCCCCU” (Figure 1B). 

Bicistronic reporter RNAs were incubated in Sf-21 insect cell lysate or rabbit reticulocyte lysate (RRL) and monitored luciferase activities and expression by incorporation of [^35^S]-methionine/cysteine (Figure 1C,D). FLuc activity was detected in Sf21 and RRL reactions with the bicistronic reporter containing the wild-type BADV-2 -1 FS. Moreover, a fusion RLuc-FLuc protein product (~100 kDa) was detected, supporting that -1 frameshifting occurred (Figure 1C,D, lane 2). An RLuc protein was also detected, indicating that a fraction of translating ribosomes did not undergo frameshifting (Figure 1C,D, lane 2). As expected, the in-frame control reporter RNA resulted in robust expression of the 100 kDa RLuc-FLuc fusion protein and loss of RLuc expression (Figure 1C,D, lane 3). Based on the relative luciferase expression, the BADV-2 -1 FS signal activity occurred at 4 and 8% efficiency in Sf-21 and RRL, respectively. Mutant bicistronic RNAs containing either a premature stop codon at the end of RLuc (Figure 1C,D, lane 1) or the slippery sequence mutant (SS Mut, UUU3783-5CCC) (Figure 1C,D, lane 4) abolished FLuc expression and activity, thus supporting the presence of a -1 FS signal. These results demonstrated that the BaDV-2 genome contains a bona fide -1 FS signal.

### 3.2. Characterization of the BADV-2 -1 FS Signal

The proposed BaDV-2 -1 FS signal contains a putative pseudoknot structure seven nucleotides downstream of the slippery sequence [31]. To delineate the minimal BADV-2 -1 FS signal, systematic 3-end deletions were created (Figure 2A). Progressive 3′ deletions significantly inhibited -1 FS activity, including 3′DEL 3 and 3′DEL 2 (Figure 2). These results suggested that the elements downstream of the putative -1 FS element may promote frameshifting.

To determine whether the predicted stem-loop and pseudoknot are important for BaDV-2 -1 FS, we generated mutations that would disrupt base-pairing and tested-1 FS activity in RRL (Figure 2C). Mutations that disrupt the bottom base-paired domain of the stem-loop (M1, M2) abolished -1 FS activity to the same extent as the SS MUT; however, compensatory mutations (M1+M2) that restored base-pairing did not rescue activity completely, suggesting that the nucleotide identities may be important (Figure 2D). Mutations that disrupt the upper base-paired region (M4) inhibited -1 FS activity; however, the corresponding disrupted base-paired region (M5) had only a minor effect (~84% of the wild type) (Figure 2D). Compensatory mutations (M4 + M5) rescued -1 FS activity to wild-type or higher levels, suggesting that nucleotide identity of ACC (nt 3804 to 3806) may be important (Figure 2D). Of note, mutant (M5), which would disrupt the predicted pseudoknot base-pairing, had only a minor effect on -1 FS activity, thus indicating that the putative pseudoknot may not be critical for frameshifting (Figure 2D). Mutations that disrupt the apical loop (M3) also did not affect -1 FS activity (Figure 2D). In summary, we have demonstrated that the BaDV-2 genome contains a bona fide -1 FS signal and identified several key nucleotides and a stem-loop that are required for activity.

### 3.3. BaDV-2 IGR Supports Internal Ribosome Entry

We investigated whether the BaDV-2 IGR supports IRES activity. Our initial analysis predicted an atypical RNA structure that had features that were similar to known dicistrovirus IRES structures [16]. The predicted start site based on the proposed IRES structure starts at a GCG codon (nucleotide 5668 to 5670). We cloned the IGR of BaDV-2 (nucleotide 5325 to 5694) into the intergenic region of the RLuc-FLuc bicistronic reporter construct (Figure 3A). IRES activity was calculated as a ratio of FLuc/RLuc, where RLuc serves as a normalization control across experiments. We incubated in vitro transcribed RNAs into Sf-21 extracts and monitored scanning-dependent RLuc translation and IRES-mediated FLuc translation. As expected, CrPV IGR IRES directed FLuc expression and activity, whereas a mutant CrPV IGR IRES that disrupts PKI base-pairing (CC6214-5 to GG mutant) abolished IRES activity (Figure 3A). The BaDV-2 IGR also resulted in FLuc expression, ~1.5-fold translation compared to the wild-type CrPV IGR IRES (Figure 3A). To further confirm that the BaDV-2 IGR can support IRES activity and that the IRES is not linked to translation of RLuc, we added a strong hairpin loop within the 5′UTR upstream of RLuc to abolish scanning-dependent translation (Figure 3B). We also inserted the strong hairpin within the 5′UTR upstream of a BaDV-2 IRES-FLuc monocistronic reporter (Figure 3C). In reactions containing the bicistronic RNAs with the 5′UTR hairpin loop, only FLuc, but not RLuc, was translated, indicating the hairpin blocks’ scanning-dependent translation of RLuc (Figure 3B,C). Further, addition of an upstream hairpin within the BaDV-2 IRES monocistronic RNA still resulted in FLuc activity. These results demonstrated conclusively that the BaDV-2 IGR contains an IRES.

### 3.4. “Mimimal” BADV-2 IGR That Directs IRES Activity

The initial predicted BaDV-2 IRES structure adopts several stem-loop and pseudoknot structures that resembled the dicistrovirus IGR IRES such as the CrPV IGR IRES [16]. A series of mutations was generated that disrupt predicted base-pairing, and BaDV-2 IRES activity was examined in vitro (Appendix A). In summary, the majority of mutants did not affect BaDV-2 IRES activity (Appendix A), suggesting an alternative RNA structure for BaDV-2 IRES translation initiation. 

We next generated a series of 5′ and 3′ deletion mutations within the IGR region to pinpoint the minimal sequence for BaDV-2 IRES translation. Initially, nucleotides 5325 to 5694 of BaDV-2 was cloned and showed IRES activity, which we denote as wild type (full-length) (Figure 4). Compared to the wild-type BaDV-2 IRES, 5′ deletions from nucleotide 5325–5421 (5′DEL 2) but not from nucleotide (nt) 5325–5366 (5′Del 1) inhibited IRES activity (Figure 4A), thus setting the boundaries from the 5′end of the BaDV-2 IRES. Using a similar approach from the 3′end, the majority of 3′ deletions from 5506–5694 (3′DEL 1-5) resulted in IRES activity as similar to or higher as compared to the wild-type IRES activity (Figure 4B). The only exception was 3′DEL 3, which showed ~30% of wild-type IRES activity (Figure 4B). To further pinpoint the minimal IRES, we generated combinations of 5′ and 3′ deletions. Combining 5′DEL 1 and 3′DEL 4 or 3′DEL 5 still showed IRES activity, whereas combining 5′DEL 2 and 3′DEL 4 inhibited IRES activity to ~40% of wild-type IRES activity (Figure 4B). Thus, the deletion analysis showed that the “minimal” BaDV-2 IRES is contained within a 140-nucleotide region from nt 5366–5506.

### 3.5. Structural Analysis of the BaDV-2 “Minimal” IRES

Small-angle X-Ray scattering (SAXS) is a method to resolve the size and shape of monodispersed macromolecules [32]. A prerequisite for SAXS is a pure, monodispersed sample that displays similar hydrodynamic properties [33]. We first purified BaDV-2 “minimal” IRES RNA made from T7 promoter by FPLC Superdex 200 (S200). The elution profile of the RNAs is as shown in Figure 5A for pure monomeric BaDV-2 IRES RNA collected at approximately 12 to 13 mL. Purified RNA was analyzed on SEC-MALS to determine the absolute molecular weight. SEC-MALS confirmed BaDV-2 IGR IRES had a molecular weight of 47.53 (±0.79%) kDa (Figure 5B), consistent with the predicted molecular weight (49.22 kDa) and indicating it is a monomer in solution. Next, we assessed BaDV-2 IRES structural envelope using SAXS. The absence of aggregation in the solution is indicated by the linearity observed in the Guinier plots (Figure 5C). The low angle region can be used to calculate Rg at inverted space, which is equal to 49.63 Å. The Guinier plot provided reciprocal information, which was then converted into a real-space electron paired distribution using indirect Fourier transformations, resulting in the derivation of the P(r) plot (Figure 5D). The P(r) plot indicates that BaDV-2 “minimal” IRES assumes an elongated structure, with a Dmax of approximately 160 Å and a real-space Rg calculated at 49.87 ± 0.22 Å, as observed in the Gaussian distribution. Here, we showed a representative model in Figure 5C, in which the light blue is the SAXS low-resolution structure, and the superimposed model is the lowest energy atomistic structure determined; the corresponding secondary structure model in represented in Figure 5E. 

To examine the secondary structure predicted by the SAXS IRES model, we re-examined the IRES translational activities by generating new mutations at key positions that would be predicted to alter base-pairing (Figure 5F, bottom). 

Mutation of UAGU to GUUA (M4), which would disrupt an apical stem-loop reduced IRES translation by ~40%, whereas all other mutations that disrupted other predicted stem-loops did not disrupt IRES translation. 

To further examine the structure of the BADV-2 IRES, we performed SHAPE structural probing (SHAPE-MaP) (Appendix A). SHAPE reactivities of each nucleotide was mapped onto the BADV-2 IRES. In general, the majority of nucleotides showed high reactivities throughout the IRES. The UAGU (nt 5447–5450), which when mutated to GUUA (M4) (Figure 5, bottom) leads to a decrease in translation by 40%, showed high SHAPE reactivity (Appendix A). In summary, the mutational analyses and structural probing analyses strongly hinted that the BaDV-2 IRES may adopt an unstructured RNA element that directs translation.

### 3.6. BaDV-2 IGR Does Not Support IRES Activity in S2 Cells

Next, we examined whether the BaDV-2 IGR IRES functions in mock- and dicistrovirus-infected S2 cells. We tested in both mock and infected cells as it has been shown that IRES translation is stimulated under virus infection [34]. We transfected in vitro transcribed bicistronic RNAs containing wild-type or mutant CrPV IRES or full-length wild-type, “minimal” or the 3′deletion (3′ DEL 5) mutant BaDV-2 IRES) into S2 cells, then infected with CrPV (MOI = 20) and measured FLuc activity at 6 h after transfection. In CrPV-infected cells, only the reporter RNAs containing the CrPV IRES displayed FLuc activity, whereas reporter RNAs containing the mutant CrPV IRES or any of the BaDV-2 IRES did not, suggesting that the BaDV-2 IRES translation is not supported in S2 cells. (Figure 6A).

We reasoned that the BaDV-2 IRES may only be functional under a viral genome context. We previously showed that the heterologous dicistrovirus IRESs can support IRES translation and infection using a chimeric CrPV infectious clone [18,35]. Briefly, we replaced the CrPV IGR IRES with the BaDV-2 IGR IRES (full length, nt 5325 to 5694) in the CrPV infectious clone (Figure 6B). We also generated a clone with an inserted P2A “stop-go” peptide site between BaDV-2 IGR IRES and ORF2 to ensure proper expression of CrPV ORF2 [36]. We first determined whether the BaDV-2 IRES could support translation of the infectious clone in Sf-21 extracts. Our results showed that the CrPV but not BaDV-2 IGR IRES supported translation of ORF2 proteins, as evidenced of VP2 expression (Figure 6C). To determine whether the CrPV-BaDV-2 chimeric clone is infectious, we transfected in vitro transcribed infectious clone RNAs into S2 cells and then followed VP2 expression by immunoblotting at 144 h after transfection. We previously showed that detection of VP2 after transfection of CrPV infectious clone RNA reflects productive CrPV infection in S2 cells [17]. VP2 was detected in S2 cells transfected with wild-type CrPV clone but not a mutant CrPV clone containing a stop codon in ORF1 or with the CrPV-BaDV-2 IGR IRES chimeric clone (both with and without P2A peptide site) (Figure 6D). Moreover, cytopathic effects were observed in S2 cells transfected with the wild-type CrPV clone but not with the other clones. In summary, the BaDV-2 IGR IRES is not active in *Drosophila* cells or under CrPV infection.

### 3.7. BaDV-2 IGR IRES Does Not Bind to Purified Ribosomes

A unique feature of dicistrovirus IGR IRESs is its ability to recruit ribosomes directly without the aid of initiation factors [37]. The CrPV IGR IRES can assemble ribosomes by first recruiting 40S subunits followed by 60S subunit joining; however, ~8% of the time, the CrPV IRES can also bind to pre-formed 80S ribosomes [8,38]. We assessed ribosomal binding to the IRES using an established filter binding assay [35]. Incubating increasing concentration of purified salt-washed ribosomes with wild-type CrPV IRES resulted in CrPV IRES: ribosome complexes with an apparent K_D_ of 1.4 ± 0.3 nM (Figure 7), which is similar to previous reports [35]. By contrast, a mutant CrPV IRES containing disruptions that disrupt all three pseudoknot structures showed little to no 80S binding (Figure 7). The BaDV-2 IRES did not bind to purified ribosomes (Figure 7, K_D_ = 940.3 ± 4377 nM), thus indicating that this IRES is likely not using a factorless mechanism for translation. 

### 3.8. BaDV-2 IGR IRES Initiates Translation from an AUG Start Codon

Our results so far point to an atypical IRES mechanism that drives BaDV-2 IGR IRES translation. We next focused on determining the start site of the BaDV-2 IGR IRES. In-frame stop codons (SC) were inserted across the full-length BaDV-2 IGR (Figure 8A). Stop codon insertions at nt 5503, 5608, and 5668 (SC1, 2, 3) all decreased IRES activity to ~5–25% of wild-type IRES activity (Figure 8, thus suggesting that the start site is upstream of nt 5503. Moreover, these experiments are in line with the deletion analysis that the “minimal” IRES is contained between nt 5366 and 5506. A stop codon insertion at nt 5444 (SC4) did not affect IRES activity, thus indicating that the start codon is between nts 5444 and 5503. We hypothesized that the AUG (nt 5491–5493) codon may be the initiation codon of the IRES. We mutated the AUG (nt 5491–5493) codon to UAG (M1) or GCG (M2) or deleted the AUG (DEL). Deletion of the AUG codon or mutating to UAG (M1) dramatically inhibited IRES translation by ~75–80% compared to wild-type IRES activity (Figure 8), albeit some residual ~20–25% IRES activity was observed. We reasoned that the residual IRES activities with the AUG codon mutations may be due to ribosomes recruited to the BaDV-2 IRES and then scanned downstream to initiate translation from the AUG codon of FLuc. To address this, we deleted the AUG codon of FLuc in combination with the deletion of the mutant AUG (nt 5491–5493) codons. Deleting the AUG codon of FLuc within the wild-type BaDV-2 IGR IRES inhibited IRES activity by ~10–15% compared to the wild-type IRES with the AUG codon of FLuc. By contrast, combining the deleted AUG codon of FLuc with deletion of AUG (nt 5491–5493) or M1 or M2 mutants abolished IRES-mediated FLuc activity completely. These results strongly demonstrated that the BaDV-2 IGR IRES initiates from the AUG (nt 5491–5493) codon and that a fraction of the 40S ribosomal subunits recruited to the IRES can scan to initiate translation from a downstream AUG codon. 

### 3.9. BaDV-2 IGR IRES Translation Requires eIF4A and eIF2

To determine whether the BaDV-2 IRES uses translation factors, we tested IRES translation in extract incubated with inhibitors that target specific translation initiation factors (Figure 9A). Specifically, hippuristanol inhibits eIF4A helicase activity and NSC119889 blocks eIF2 from binding to initiator Met-tRNA_i_ [39,40]. We pre-incubated these inhibitors in RRL or Sf-21 extracts followed by addition of IRES-containing reporter RNAs. Incubating with hippuristanol (10 μM and 15 μM) or NSC119889 (12.5 μM) significantly decreased scanning-dependent RLuc translation, as expected (Figure 9B,C). By contrast, CrPV IRES-mediated FLuc translation was relatively resistant to hippuristanol or NSC119889, in line with the CrPV IRES driving factorless translation (Figure 9B,C) [41]. Interestingly, BaDV-2 IRES translation was sensitive to both drug treatments inhibiting translation as much as scanning dependent, thus demonstrating that eIF4A and eIF2 ternary complexes are required for BaDV-2 IRES translation. 

## 4. Discussion

To date, the dicistrovirus family contains the most streamlined mechanisms of eukaryotic translation utilizing an RNA structure to directly recruit ribosomes and initiate translation. Recently, variations in dicistrovirus IRESs have come to light, showing that the dicistrovirus IRESs can have distinct ribosome assembly pathways (ex. IRES binds to preformed 80S) and can initiate from ribosomal P or A sites [6,7]. Here, in this study, we focused on a mechanism of a whitefly dicistrovirus virus, BaDV-2. Like other dicistroviruses, the BaDV-2 intergenic region can support IRES translation in vitro; however, several evidence demonstrate that this mechanism is distinct from other known dicistrovirus IRES mechanisms. One, the BaDV-2 IRES does not bind to ribosomes directly and likely requires a subset of translation initiation factors and other factors. Our results showed that eIF2 and eIF4A are required for BaDV-2 IRES translation. Two, the BaDV-2 IRES is within a 140-nucleotide element and starts translation from an AUG codon, in line with the requirement of eIF2 for IRES translation. Three, the BaDV-2 IRES likely does not utilize an RNA structure or may adopt different conformations to direct ribosome recruitment. Extensive mutational analysis and SAXS analysis and the requirement of the RNA helicase eIF4A support this model. Our results demonstrate that the BaDV-2 contains a novel IRES mechanism that is distinct from other dicistrovirus IRES mechanisms to date and may represent a novel class of IRES.

Our studies highlight the diverse IRES mechanisms that may be used within the family *Dicistroviridae*. It is likely that the BaDV-2 IRES evolved mechanisms that require factors in order to adapt to and drive viral protein synthesis in BaDV-2 infection in whiteflies. Thus, there may be specific cellular infection conditions that promote BaDV-2 IRES translation. Indeed, we showed that the BaDV-2 IRES is not functional in *Drosophila* cells and under CrPV infection (Figure 6). However, it is interesting to note that the BADV-2 IRES translation is supported in vitro in Sf21 extracts and not in *Drosophila* cells, thus pointing to specific host factors required for BADV-2 IRES translation. To date, until this study, dicistrovirus IRES mechanisms in general all utilize a factorless ribosome assembly mechanism; however, it is clear that the diverse translation mechanisms are utilized under specific dicistrovirus infections. The BaDV-2 IRES represents one of the most divergent IRES mechanisms within the family *Dicistroviridae*, which likely means that there are many alternative viral translation strategies that are utilized by dicistroviruses. It will be interesting to follow the evolutionary trajectory of dicistroviruses and whether they utilize a factorless IRES mechanism, a BaDV-2 IRES-like mechanism, or a yet undiscovered translation mechanism. 

The BaDV-2 IRES utilizes at least eIF2 and eIF4A for IRES translation. Moreover, the exact structure of BaDV-2 that supports IRES function needs further investigation. Although SAXS analysis predicted a structure containing several stem-loops, our mutational analysis and SHAPE structural probing did not support this model. The data point to an unstructured or potential remodeling of the RNA structure that supports BaDV-2 IRES function, which is supported by its requirement for eIF4A. This evidence suggests a mechanism that is a bit reminiscent of some cellular IRESs that are thought to use unstructured elements that may be ribosome landing elements [42,43]. Further investigation into the specific BaDV-2 IRES elements and/or RNA conformations is warranted.

Our results have for the first time provided biochemical evidence of a bona fide -1 FS signal within a dicistrovirus genome. The BaDV-2 -1 FS signal contains standard elements of a frameshift element including a stem-loop and an upstream slippery sequence [28]. We ruled out the requirement of a predicted pseudoknot for BaDV-2 -1 FS activity in vitro. The -1 FS signal is between the 3C protease and RNA-dependent RNA polymerase (RdRP) [16]. Thus, it is likely that the -1 FS signal regulates the exact stoichiometry of the RdRP to other viral non-structural proteins for precise replication activity during the viral life cycle. It will be of interest to identify other -1 FS signals within discistroviruses and whether this is more widespread within this viral family. It is noteworthy that BaDV-1 has a typical dicistronic genome, while BaDV-2 is primarily tricistronic.

The whitefly complex *Bemisia tabaci* is a critical agricultural pest that occurs in most tropical and subtropical countries and affects important crops, like cotton, cassava, beans, soybeans, and vegetables [44,45,46,47]. The identification of viruses such as BaDV-1 and BaDV-2 that are associated with whiteflies may provide a direction for biopesticide approaches to control these pests [16,48], for example, those baculoviruses used for controlling caterpillars [49,50,51]. Understanding the molecular mechanisms, such as IRES-mediated viral protein synthesis, will provide valuable insights into the viral infection strategies of these pest-associated viruses that may be exploited for therapeutic approaches in agriculture.

## Figures and Tables

**Figure 1 viruses-16-00695-f001:**
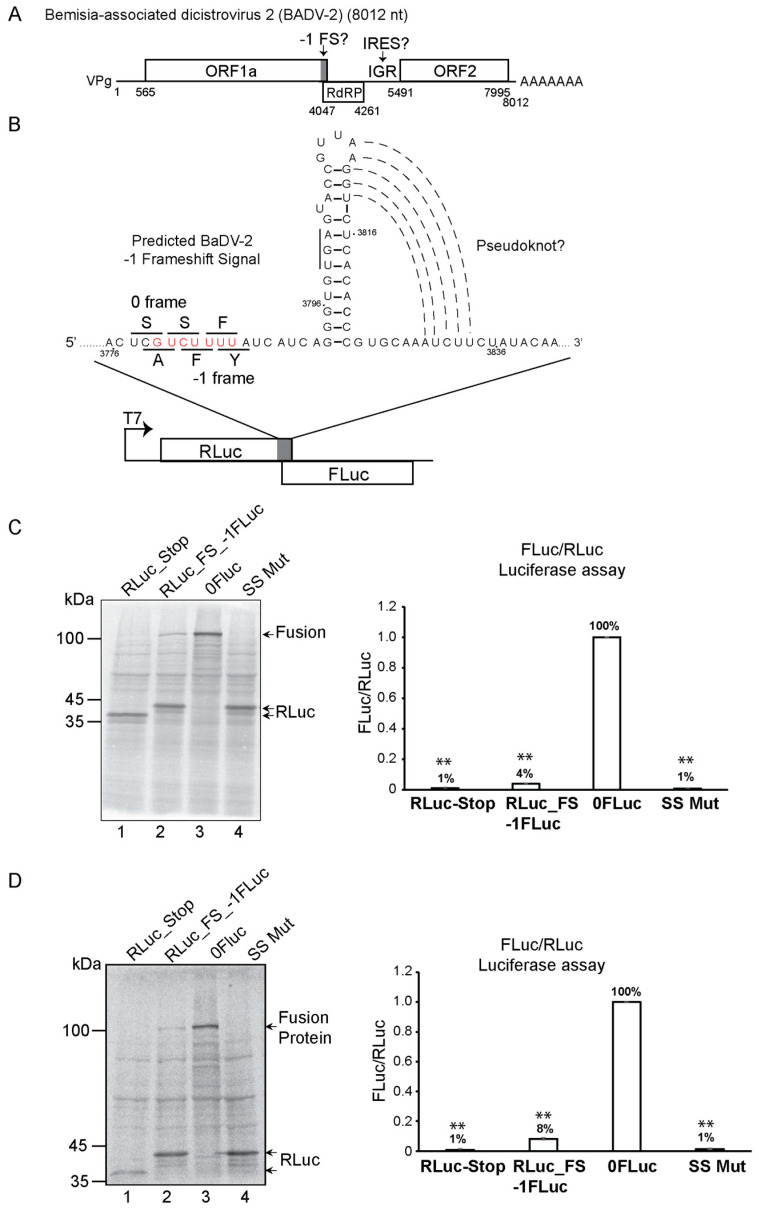
In vitro translation assays of BaDV-2 -1 FS in Sf-21 and RRL. (**A**) Simplified schematic view of the BaDV-2 genome. (**B**) The predicted secondary structure of BaDV-2 -1 frameshifting signal and schematic view of the bicistronic reporter with RLuc and FLuc. The slippery sequence is shown in red letters and the amino acids decoded in the -1 and 0 frames are shown in black letters. Bicistronic reporter RNAs containing wild-type or mutant BaDV-2 -1 FSS and in-frame controls were incubated in (**C**) Sf-21 or (**D**) RRL extracts. “RLuc Stop” contains a premature stop codon at the end of RLuc; “RLuc_FS_-1FLuc” contains the wild-type -1 FS with FLuc in -1 frame; “SS Mut” contains a mutant slippery sequence (UUU3783-5CCC); “0FLuc” is the in-frame control in which the RLuc and FLuc are both in the 0-reading frame. (left) Reactions were resolved on SDS-PAGE gel and monitored by phosphorimager analysis. Shown is a representative SDS-PAGE gel from at least three independent experiments. (right) Luciferase activities were measured and calculated as a ratio of FLuc/RLuc and normalized to the in-frame control (0FLuc). Groups were compared to 0FLuc. A one-way ANOVA statistical test was used to determine the *p* values (*** p* < 0.01). Shown are averages from at least three independent experiments ± standard deviation.

**Figure 2 viruses-16-00695-f002:**
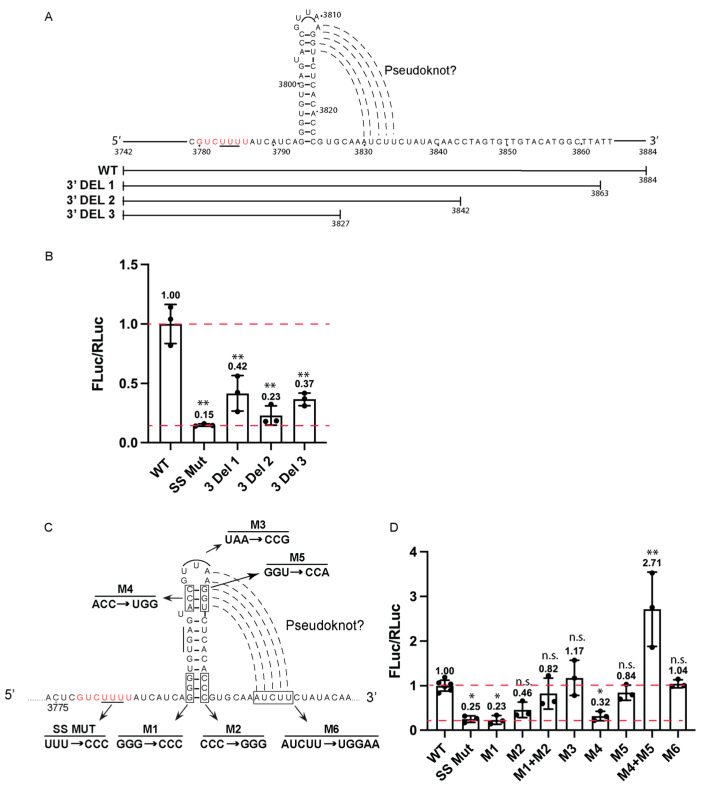
Structural analysis of BaDV-2 -1 FS element. (**A**) Schematic of deletion mutants surrounding the putative BaDV-2 -1 FS. The slippery sequence is shown in red letters. (**B**) In vitro transcribed RNAs were incubated in RRL and luciferase activities were measured and normalized to wild-type BaDV-2 -1 FS (BaDV-2 FS WT). Groups were compared to the wild-type BaDV-2 -1 FS. (**C**) Schematic view of BaDV-2 -1 FS mutations. (**D**) In vitro transcribed RNAs were incubated in RRL, and luciferase activities were measured and normalized to wild-type BaDV-2 -1 FSS (BaDV-2 WT). The top red dotted line represents Fluc/RLuc ratio of the wild-type -1 FS activity normalized to 1. The lower red dot line represents the background characterized by a mutation in the slippery site (SS MUT, UUU3783-5CCC) that abolishes -1 FS activity. Groups were compared to BaDV-2 WT. A one-way ANOVA statistical test was used to determine the *p* values (** p* < 0.05, *** p* < 0.01). “n.s.” denotes the difference is not significant between the experimental groups and WT (*p* > 0.05). Shown are averages from at least three independent experiments ± standard deviation.

**Figure 3 viruses-16-00695-f003:**
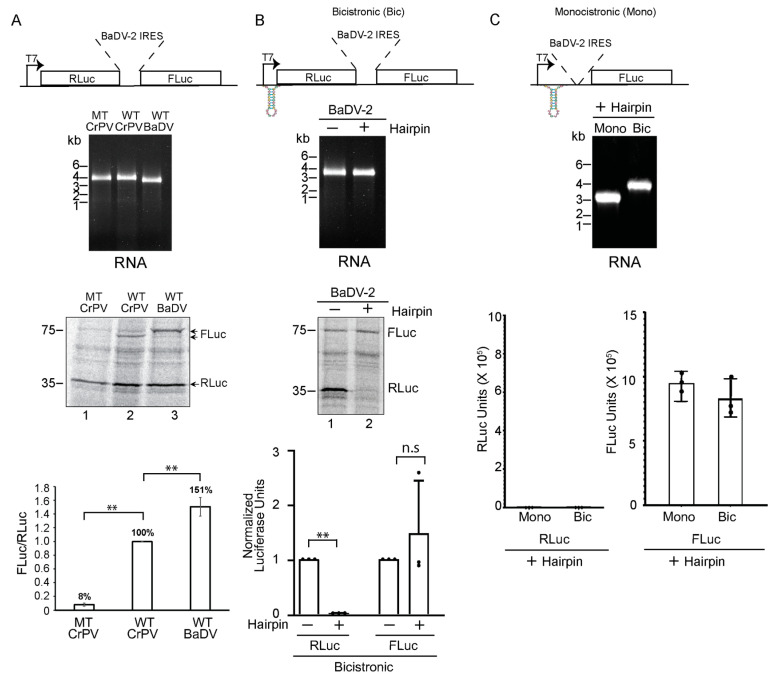
BaDV-2 IGR supports internal ribosome entry. (**A**) Bicistronic reporter RNAs containing either the CrPV IGR or the BaDV-2 IGR, (**B**) bicistronic RNAs with a 5′UTR strong hairpin, or (**C**) monocistronic reporter RNAs with a 5′UTR hairpin and the BaDV-2 IGR (top schematics) were incubated in Sf-21 extracts. In vitro transcribed RNAs were analyzed by agarose gel analysis (middle). Translation of FLuc and RLuc was measured by either radioisotope [^35^S]-methionine/cysteine incorporation followed by SDS-PAGE and phosphorimager analysis or luciferase assays (graphs shown bottom). A paired *t*-test was used to determine the *p* values. *** p* < 0.01. “n.s.” denotes the difference is not significant between the control groups and the experimental groups (*p* > 0.05). Shown are representative SDS-PAGE gels and the averages from at least three independent experiments ± standard deviation.

**Figure 4 viruses-16-00695-f004:**
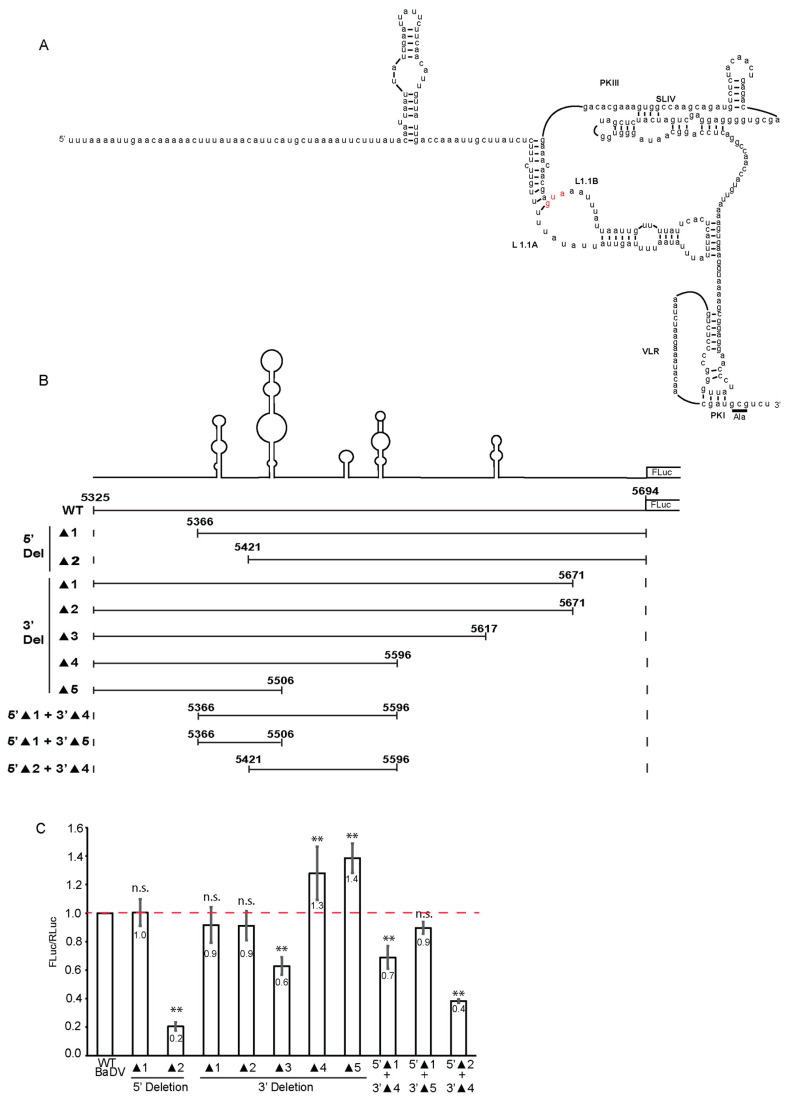
Deletion analysis of BaDV-2 IGR IRES. (**A**) Schematic of the initial predicted structure of the BaDV-2 IGR IRES. The red AUG denotes the predicted start codon. (**B**) 5′ and 3′ deletions were generated within the BaDV-2 IGR. “▲” = DEL = deletion. (**C**) Bicistronic RNAs containing the corresponding deletion mutants were incubated in Sf-21 extracts and RLuc and FLuc activities measured and normalized to the wild-type BaDV-2 IRES. The red line denotes the FLuc:RLuc ratio of the wild type IRES for comparison. A one-way ANOVA statistical test was used to determine the *p* value and thus the significance levels. Groups were compared to WT BaDV. *** p* < 0.01. “n.s.” denotes the difference is not significant between the experimental groups and WT (*p* > 0.05). Shown are the averages from at least three independent experiments ± standard deviation.

**Figure 5 viruses-16-00695-f005:**
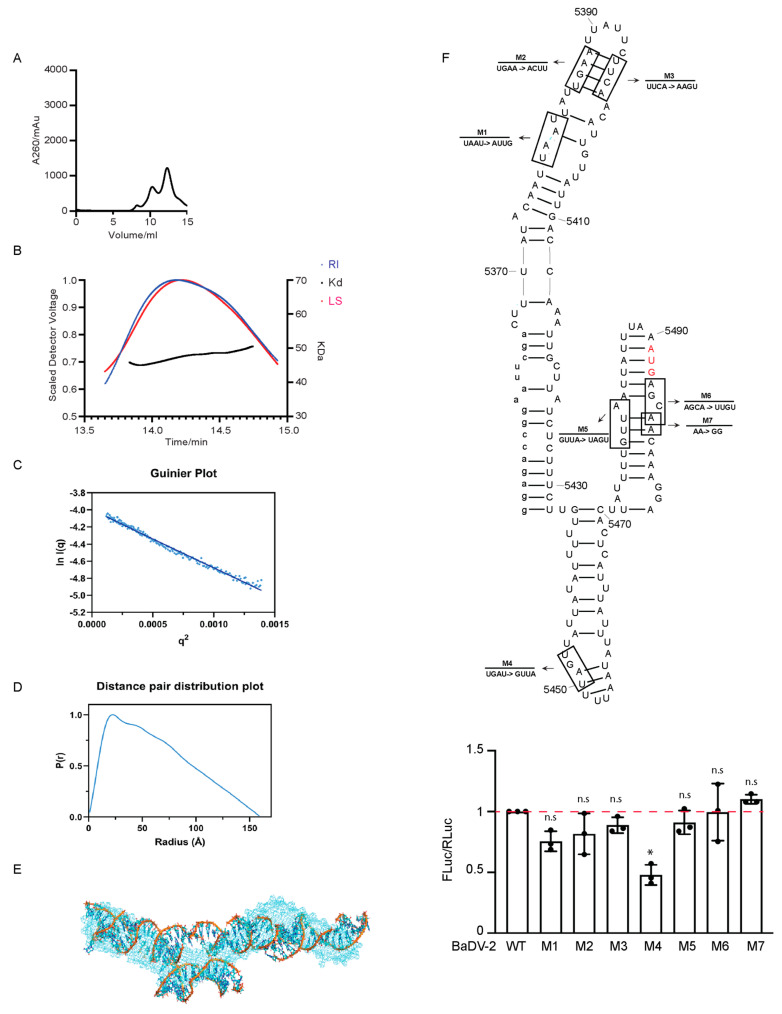
Structural analysis of BaDV-2 IGR IRES. (**A**) FPLC profiles associated with BaDV-2 IGR “minimal” IRES IVT RNA purification. (**B**) SEC-MALS traces of the peak from the BaDV-2 IRES RNA run and the absolute molecular weight across them. (**C**) The absence of aggregation and the presence of a liner Guinier region in the sample indicates its purity, which enables the calculation of Rg from the low-angle region data and signifies the homogeneity of the sample. (**D**) Normalized pair distance distribution plots for BaDV-2 IRES RNA permit the determination of Rg derived from the SAXS dataset and including each molecule’s Dmax. P(r) versus radial distance plot indicates an elongated RNA structure in solution. (**E**) Predicted BaDV-2 IRES RNA structure model from SAXS data (light blue) overlapped with atomistic structure, represented in ribbons. (**F**) The secondary structural model of BaDV-2 IRES corresponding to the predicted model as shown in (**E**). The red letters denote the predicted AUG start codon. Mutation analysis of BaDV-2 “minimal” IRES with bicistronic reporter Sf-21 translation assays were performed and the results (M1-7) were measured by luciferase assays and normalized to wild-type (WT) BaDV-2 IGR IRES “minimal” IRES (bottom). Groups were compared to WT BaDV-2 IRES (red dotted line). An ANOVA statistical test was used to determine the *p* value and thus the significance levels. ** p* < 0.05. “n.s.” denotes the difference is not significant between the experimental groups and WT (*p* > 0.05). Shown are the averages from at least three independent experiments ± standard deviation.

**Figure 6 viruses-16-00695-f006:**
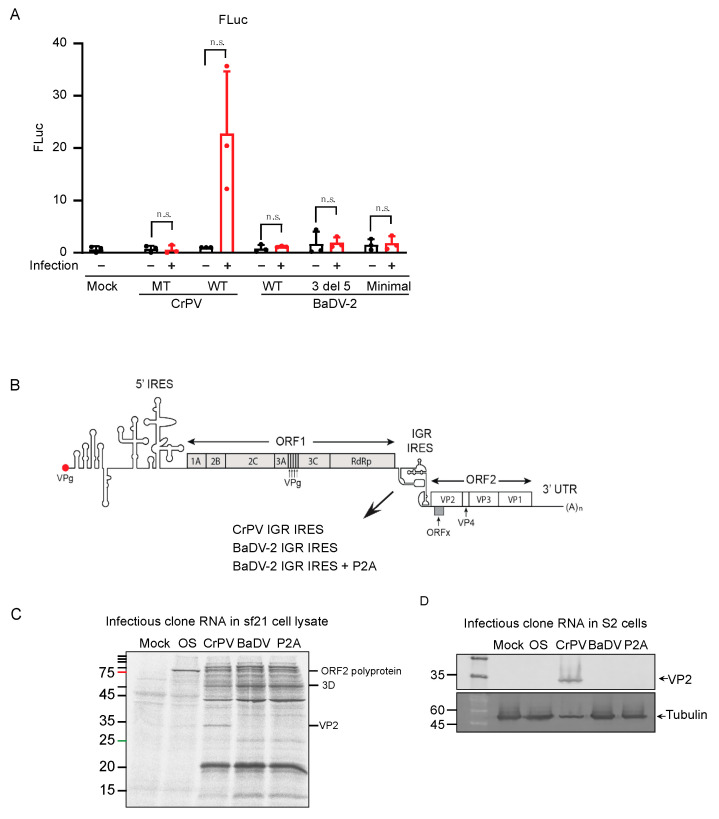
BaDV-2 IGR IRES activity in infectious clone and in cells. (**A**) Bicistronic reporters were transfected into S2 cells followed by mock infection or CrPV infection (MOI = 20). Cells were collected 6 h post transfection. Bicistronic RNAs tested contain the wild-type CrPV, mutant CrPV (CC6214-5 to GG to disrupt PKI base-pairing), wild-type BaDV-2 full-length IGR IRES (WT BaDV-2), BaDV-2 3′ DEL 5, and BaDV “minimal” IRES (double deletion at both 3′ and 5′ end. Luciferase activities were measured and normalized to wild-type CrPV with mock infection. Red and black dots represent data points from three independent experiments. A paired *t*-test was used to determine the *p* value and thus the significance levels. “n.s.” denotes the difference is not significant between the control groups and the experimental groups (*p* > 0.05). Shown are the averages from at least three independent experiments ± standard deviation. (**B**) Schematic of infectious clone CrPV (CrPV, CrPV_IGR_IRES) and chimeric clones with CrPV IGR IRES replaced with BaDV-2 IGR IRES (BaDV-2 IGR IRES) and with P2A-site added after BaDV-2 IGR IRES (BaDV-2 IGR IRES+ P2A). (**C**) Infectious clone RNAs were incubated in Sf-21 extracts containing [35S]-methionine/cysteine and then monitored by SDS-PAGE analysis. Mock = mock transfection; OS = ORF1Stop; CrPV = pCrPV; BaDV = BaDV-2 IGR IRES; P2A = BaDV-2 IGR IRES + P2A. Shown is a representative gel from at least three independent experiments. (**D**) In vitro transcribed infectious clone RNAs were transfected into S2 cells for 144 h. VP2 expression was detected by immunoblotting. Shown are a representative SDS PAGE gel and the averages from at least three independent experiments ± standard deviation.

**Figure 7 viruses-16-00695-f007:**
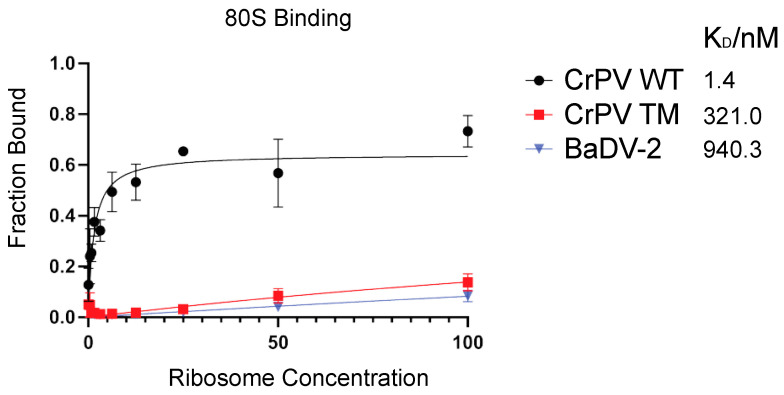
Affinity of 80S-BaDV-2 IGR IRES complexes. Filter binding assays. [32P]-BaDV-2 IGR IRES, [32P]-(CrPV) IGR IRES or mutant (∆PKI/II/III) CrPV IGR IRES were incubated with increasing amounts of purified salt-washed 80S. The fractions bound were quantified by filter binding assay followed by phosphorimager analysis. CrPV WT has a K_D_ value of 1.4 nM, CrPV mutant (MT, mutations that disrupt all three pseudoknot base-pairings) has a K_D_ value of 321.0 nM and WT BaDV-2 has a K_D_ value of 940.3 nM. Shown are the averages from at least three independent experiments ± standard deviation.

**Figure 8 viruses-16-00695-f008:**
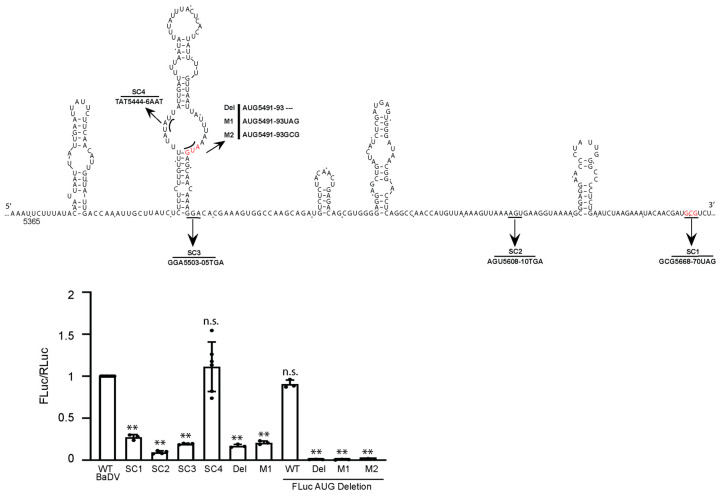
Stop codon mutations to confirm the BaDV-2 IGR IRES initiation site. (**A**) Schematic of stop codon mutations along the BaDV-2 IGR IRES (upper). The red letters denote the predicted AUG start codon. In vitro translation assays were performed in Sf-21 cell lysate with the indicated mutations using bicistronic reporter RNAs. RLuc and FLuc activities were measured and normalized to the wild-type BaDV-2 bicistronic RNA (bottom). An ANOVA statistical test was used to determine the *p* value and thus the significance levels. Groups were compared to WT BaDV. ** represents *p* < 0.01. “n.s.” denotes the difference is not significant between the experimental groups and WT (*p* > 0.05). Shown are the averages from at least three independent experiments ± standard deviation.

**Figure 9 viruses-16-00695-f009:**
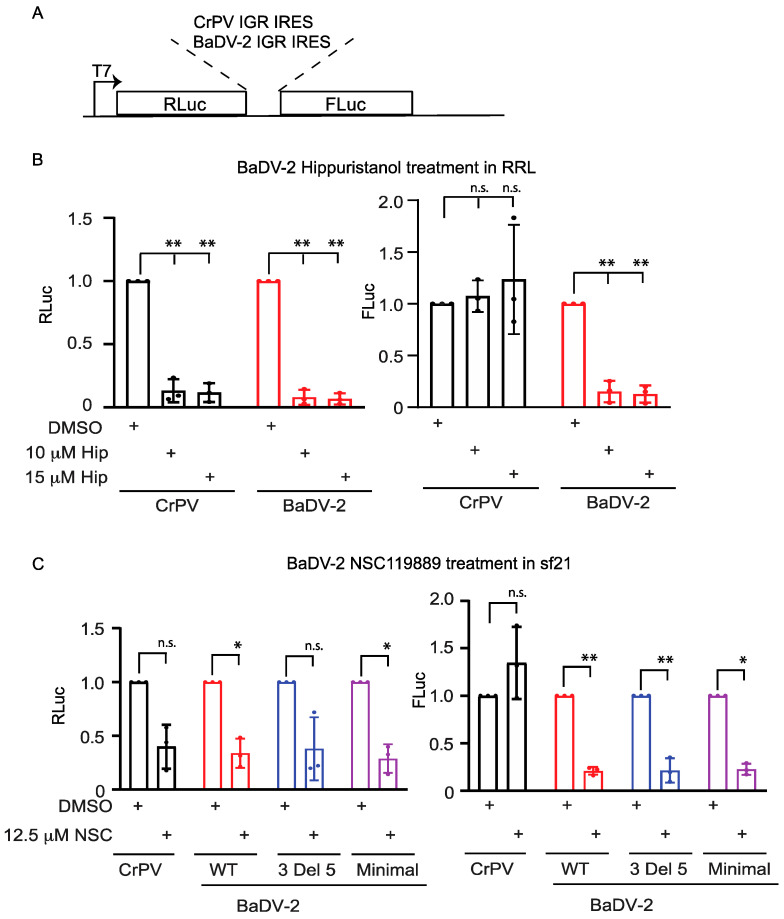
Effects of translation inhibitors on BaDV-2 IRES translation. (**A**) Schematic illustration of bicistronic reporter RNAs with CrPV or BaDV-2 IGR IRESs. In vitro transcribed reporter RNAs were incubated (**B**) in RRL with hippuristanol or (**C**) in Sf21 extracts with NSC119889 at the indicated concentrations. RLuc and FLuc activities were measured and normalized to DMSO-treated lysates with wild-type CrPV or BaDV-2 IGR IRES bicistronic RNAs. A one-way ANOVA statistical test was used to determine the *p* values. ** p* < 0.05, *** p* < 0.01. “n.s.” denotes the difference is not significant between the control groups and the experimental groups (*p* > 0.05). Shown are the averages from at least three independent experiments ± standard deviation.

## Data Availability

The original contributions presented in the study are included in the article/Appendix A, further inquiries can be directed to the corresponding author/s.

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
