# Peer review of "Factor-Dependent Internal Ribosome Entry Site and -1 Programmed Frameshifting Signal in the Bemisia-Associated Dicistrovirus 2"

_viruses, 2024, doi:10.3390/v16050695_

Round 1
Reviewer 1 Report
Comments and Suggestions for Authors
Chen et al report a comprehensive analysis if the intergenic region of Bemisia-associated dicistrovirus 2. The paper is split into two parts, characterization of the frame shift element and characterization of a potential IRES activity. The frame shift element work is clear and well presented, is of general interest. Some suggested experiments for improvement are outlined below. The IRES element of the paper is less well presented, somewhat confused and needs some other controls given the important nature of the finding of an unusual class of IRES.
Major comments:
1) The authors could use SHAPE mapping or a similar structural probing technique to confirm the structure of the frame shifting stimulatory element and lack of requirement for the pseudoknot. Is it possible that the pseudoknot could have an inhibitory role in certain circumstances given the surprising increase in frameshifting seen for the M4+M5 double mutant in Figure 2?
2) The authors should test a control monocistronic RNA in which a stable stem is positioned 5' to the tested RNA sequence. Given the strong nature of the claims in the paper of an IGR IRES of novel/no structure that requires eIF4A and eIF2, this control would help to deal with concerns that the detected IRES activity is from partially degraded RNA that would still be translated, likely with a role for 4A and eIF2, even with the current control of a stem loop upstream of the Rluc orf. Alternatively, a stem loop could be introduced between the first orf and the IGR region in the bicistronic construct.
3) Why do the authors swap from Sf21 cell lysates for S2 cells for transfections? The change just happens in the text, some justification is needed to explain why. Why not use Sf21 cells for transfection? Or why not use S2 lysates for in vitro translations? The lack of evidence for IRES activity in cells is a major concern and these seem like important controls.
4) The authors show Fluc/Rluc ratios for all the in vitro data but only fluc data for the in cell experiments. The Rluc data should also be included.
5) The authors should include another mutant in the experiment in Figure 8, introducing one or more AUGs upstream of AUG 5491-5393, in the context of the AUG5491/FlucAUG double mutant to identify how far upstream ribosomes can initiate translation in this sequence.
6) While I understand that teh authors are moving between different potential structures as no RNA structure mapping is reported, jumping between the different structures is confusing and presentation should be improved. For example the highlighted nucleotides in Figure 4 could be similarly colored in Figure 5. In Figure 5 the correct numbers for nucleotide locations should be included to ease comparison between different figures.
7) Lines 501-505 and 523-525: in the discussion, the authors suggest that no or alternative RNA structures direst IRES activity. The only structural work done is with SAXS which clearly shows structure, but the authors then go back to a basic structure prediction in a subsequent figure, ignoring the SAXS data. The authors say that their mutational data doesn't support the structures predicted or determined in SAXS (lines 523-525), but the readout was translation activity, not structure. This should be acknowledged in the text. The paper would benefit from RNA structure probing so that claims that mutation disproves structure can be supported.
Minor comments:
Lines 341-344: It's not clear what was done here from the text. Were these all new mutations designed based on the SAXS data in figure 5 as these do not match with the mutations listed in figure S1, but the text reads as these mutations are those from Figure S1 mapped onto the SAXS structure, can this be clarified?
Line 418: Accessed/assessed
Line 386: evidenced
Figure 9 Legend: Correct phrasing: 'were incubated in B) RRL with hippuristanol or C) in Sf21 extracts with NSC119889 at the indicat-ed concentrations.'
Lines 301-302: The text states that 5'del1 and 5'del 2 inhibit IRES activity, but the data shows that only 5'del2 inhibits activity
Author Response
We thank the reviewers for their comments and suggestions. In the revised resubmission, we have addressed all comments from the reviewers, including new data in Figure 3, Figures S2 and S3 (SHAPE analysis). We have included the original comments from reviewers italicized with our response below.
We believe that the manuscript is significantly improved, and we hope that it is acceptable for publication in Viruses.
REVIEWER 1 comments are italized
RESPONSES in blue
Chen et al report a comprehensive analysis if the intergenic region of Bemisia-associated dicistrovirus 2. The paper is split into two parts, characterization of the frame shift element and characterization of a potential IRES activity. The frame shift element work is clear and well presented, is of general interest. Some suggested experiments for improvement are outlined below. The IRES element of the paper is less well presented, somewhat confused and needs some other controls given the important nature of the finding of an unusual class of IRES.
1) The authors could use SHAPE mapping or a similar structural probing technique to confirm the structure of the frame shifting stimulatory element and lack of requirement for the pseudoknot. Is it possible that the pseudoknot could have an inhibitory role in certain circumstances given the surprising increase in frameshifting seen for the M4+M5 double mutant in Figure 2?
RESPONSE: We thank the reviewer for this comment. We agree that the increase in FS observed with M4+M5 was surprising and could be due to an inhibitory effect by the potential pseudoknot. However, mutations M5 and M6 on their own did not affect the FS activity, thus ruling out an inhibitory effect by the pseudoknot. At this moment, the mutational analysis all point to a lack of a pseudoknot required for FS. As for the increase in FS activity due to M4+M5, this will require more analysis especially using SHAPE or DMS structural probing as the reviewer suggests, however, these analyses will be better suited in a more in-depth future study.
2) The authors should test a control monocistronic RNA in which a stable stem is positioned 5' to the tested RNA sequence. Given the strong nature of the claims in the paper of an IGR IRES of novel/no structure that requires eIF4A and eIF2, this control would help to deal with concerns that the detected IRES activity is from partially degraded RNA that would still be translated, likely with a role for 4A and eIF2, even with the current control of a stem loop upstream of the Rluc orf. Alternatively, a stem loop could be introduced between the first orf and the IGR region in the bicistronic construct.
RESPONSE: We thank the reviewer for this. We have provided two experiments to address this. 1) We now include representative gels showing the integrity of the RNAs in Figure 3 (please attached new Figure 3). 2) As the reviewer suggests, we have included the hairpin upstream of the BADV-2 IRES in a monocistronic reporter and demonstrate that translation of the IRES is still active and translation is as active as within the bicistronic reporter (please see attached new Figure 3).
3) Why do the authors swap from Sf21 cell lysates for S2 cells for transfections? The change just happens in the text, some justification is needed to explain why. Why not use Sf21 cells for transfection? Or why not use S2 lysates for in vitro translations? The lack of evidence for IRES activity in cells is a major concern and these seem like important controls.
RESPONSE: We thank the reviewer for these insightful comments. We agree, there may be differences in SF21 vs S2 cells that determine whether the IRES works or not. Firstly, testing in S2 extracts would make sense but we currently do not have S2 lysates for these experiments. Secondly, for testing in Sf21 cells, this is a great idea but we do not have these cells at this time. However, we did attempt to transfect the bicistronic reporter RNAs in Sf9 cells, but transfection failed (no Renilla or Firefly luciferase and we tried several transfection reagents). Further optimization of transfection in Sf9 cells will require further time and effort to resolve.
In the past, we have used these two systems (S2 vs Sf21) interchangeably and have not seen any differences. This would be the first time that we have observed these differences and thus will be examined in more detail in the future. To address this comment, we have added the sentence in the discussion on lines 621-623 “However, it is interesting to note that the BADV-2 IRES translation is supported in vitro in Sf21 extracts and not in Drosophila cells, thus pointing to specific host factors required for BADV-2 IRES translation.”
4) The authors show Fluc/Rluc ratios for all the in vitro data but only fluc data for the in cell experiments. The Rluc data should also be included.
RESPONSE: We now provide all of the data in a new Figure S2 showing all RLuc and FLuc/RLuc data.
5) The authors should include another mutant in the experiment in Figure 8, introducing one or more AUGs upstream of AUG 5491-5393, in the context of the AUG5491/FlucAUG double mutant to identify how far upstream ribosomes can initiate translation in this sequence.
RESPONSE: Thank you for the suggestion. We agree it is a good idea that see if translation can start upstream (or downstream) of the putative AUG5491. However, for the purposes of this study, we believe we have determined the start site of the IRES and thus, future analysis of placement of the AUG upstream/downstream will be conducted at a later time.
6) While I understand that the authors are moving between different potential structures as no RNA structure mapping is reported, jumping between the different structures is confusing and presentation should be improved. For example the highlighted nucleotides in Figure 4 could be similarly colored in Figure 5. In Figure 5 the correct numbers for nucleotide locations should be included to ease comparison between different figures.
RESPONSE: We have highlighted the nucleotides in Figure 5 as in Figure 4. We thank the reviewer for pointing this out and it has made the figure clearer. We have also included the initial IRES model (from Figure S1) to Figure 4 for clarification.
7) Lines 501-505 and 523-525: in the discussion, the authors suggest that no or alternative RNA structures direst IRES activity. The only structural work done is with SAXS which clearly shows structure, but the authors then go back to a basic structure prediction in a subsequent figure, ignoring the SAXS data. The authors say that their mutational data doesn't support the structures predicted or determined in SAXS (lines 523-525), but the readout was translation activity, not structure. This should be acknowledged in the text. The paper would benefit from RNA structure probing so that claims that mutation disproves structure can be supported.
RESPONSE: We agree with the reviewer. We performed SHAPE-MaP analysis and is now a new Figure S3. In general, the SHAPE analysis showed that higher reactivity throughout the IRES, thus in line that the structure determined by SAXS or by modeling is not supported. Thus, we believe that the IRES is relatively unstructured that may be important for its function.
Minor comments:
Lines 341-344: It's not clear what was done here from the text. Were these all new mutations designed based on the SAXS data in figure 5 as these do not match with the mutations listed in figure S1, but the text reads as these mutations are those from Figure S1 mapped onto the SAXS structure, can this be clarified?
RESPONSE: We thank the reviewer for the suggestion. The mutations in Figure 5 are new mutations for testing the SAXS-predicted structure but not from previous S1 mutations. We are now clarified this section, lines 397-399. “To examine whether parts of the secondary structure of SAXS IRES model is functions in translation, we re-examined the IRES translational activities by generating new mutations at key positions that would be predicted to alter basepairing (Figure 5F, bottom)."
Line 418: Accessed/assessed
Line 386: evidenced
RESPONSE: We thank the reviewer for pointing this out. Corrected.
Figure 9 Legend: Correct phrasing: 'were incubated in B) RRL with hippuristanol or C) in Sf21 extracts with NSC119889 at the indicat-ed concentrations.'
RESPONSE: We thank the reviewer for pointing this out. We now updated the Figure 9 to correct this.
Lines 301-302: The text states that 5'del1 and 5'del 2 inhibit IRES activity, but the data shows that only 5'del2 inhibits activity
RESPONSE: We thank the reviewer for pointing this out. We now updated the manuscript to “Compared to the wild-type BaDV-2 IRES, 5’ deletions from nucleotide 5325-5421 (5’DEL 2) but not from nucleotide (nt) 5325-5366 (5’Del 1) inhibited IRES activity (Figure 4A), thus setting the boundaries from the 5’end of the BaDV-2 IRES.” (lines 338-339).

Reviewer 2 Report
Comments and Suggestions for Authors
In this manuscript Chen et al., nicely describe the structure and functional characterization of a programmed -1 frameshift (-1 FS) and the intergenic (IGR) IRES in the Bemisia-associated dicistrivirus 2 (BaDV-2). The authors provide evidence that the predicted -1 FS actually promotes frameshift in vitro, using bicistronic reporter constructs. Similarly they also provide evidence for the initiation of translation driven by the IGR IRES but pointing to a molecular mechanism different from that described for other members of the dicistroviruses family. The authors combine structural and functional studies very well to show unambiguously the implication of these two “tools” of the viral genome in the regulation of the viral translation. However I have found several points that need to be considered.
Major points
1. The experiments performed and, therefore the results are difficult to be understood without a clear and detailed diagram of the BaDV-2 dicistrovirus.
A schematic representation of the genome as provided in Figure 6B, but enlarged and including all the details necessary to follow the different experiments i.e., -1 FS with the hypothetical structure features; the IRG IRES, etc, should be provided as Figure 1, somewhere in the introduction.
2. The statistical analysis performed throughout the paper, the ANNOVA test, is not adequate for such a small sample size. ANNOVA is a parametric test, but for this sample size the authors should use a non-parametric test, for example, the Mann Withney test, or any other non-parametric test.
3. Characterization of the BaDV-2 -1 FS signal.
It is difficult to follow the reasoning of the authors. First in the representation of the analyzed fragment they should indicate the coordinates of the real fragment they analyzed (3742-3884). Looking at the figure it seems that they are using a fragment of 3770-3865. The analysis of 3’ Del 1, 2 and 3 clearly indicate that just the deletion of the 21 nt at the 3’ end (3864 to 3865) results in a significant reduction of the -1 FS. But the authors do not give importance to this result. Instead they emphasize the inhibition by the Del 2 and Del 3 constructs. (page 5 line 219). The authors should discuss this issue. This result together with the very low frameshift efficiency of the full fragment, which has been quantified as only 4 and 8% depending on the extracts used, have the authors considered that nucleotides beyond 3884 might be necessary to achieve the 100% functionality of the -1 FS?
4. Figure 5A, the sequence cannot be read. Panel A of supplementary figure S1 should be included here as panel A. Then the current panel A should be panel B but replacing the current sequence representation of the analyzed fragment by a schematic line-drawing representation, clearly distinguishing the fragment corresponding to Fluc by a different line color or line shape. This scheme should include all structural and sequence details (e.g. by line colors necessary).
5. Pg. 9 line 302. The authors state that 5’ Del1 and Del 2 inhibited IRES activity but figure 4 indicates no effect of 5’ Del1.
Line 309, combining 5’Del 2 and 3’Del 4 inhibited IRES activity to ~17% of wild-type IRES activity (Figure 4B). But in Figure 4B indicates ~40%.
6. I agree with the authors that they have identified a 140 nt long fragment that supports translation initiation (IRES activity), but they cannot conclude that this is the minimal fragment or the minimal IRES. This statement should be carefully revised throughout the manuscript.
7. Page 10 lines 341-344. This is confusing, are the authors using the same mutants included in Figure S1?, if so the nomenclature M1 to M7 does not correspond to that used in Figure S1.
8. It is difficult to accept that there is no significant differences for Mutants M1, M3, M7 and even M2 and M5. As mentioned in point 2, this analysis should be done with a non-parametric test.
Minor points
1. Introduction line 36. I understand that the authors mean: RNA viruses carry genomes with limited coding capacity; or RNA virus genomes have limited coding capacity.
Viruses, nor any other organisms encode a genome. Genomes encode information, not organisms encode genomes.
2. Page 2 line 69 delete subunit
3. Page 3 line 117, Nucleic Acids are not transfected. Cells are transfected with nucleic acids.
4. Fig. 1B Nucleotide positions (numbers) should be indicated in the figure
5. Figure 1C and 1D, lane numbers should be indicated in the gel.
6. Figure 2, Identification of panels B, C and D is missing.
7. The meaning of n.s. should be indicated in all figures
8. Figure 2A Constructs 3’ DEL are referred to in the text as Del. Inconsistency of the nomenclature
9. Figure legend 2D, line 260. The red dot line corresponds to the WT.
10. Page 10, line 341. “…are functional in translation”
11. Figure 5F nucleotide positions, coordinates, should be the same than in the previous figures. This new numbering makes very difficult to identify the mutations.
12. 12 page 12 line 386 “evidence”
13. KD should be Kd
14. Figure 8, M2 has not been analyzed?
Author Response
REVIEWER 2 comments are italized
RESPONSES in blue
In this manuscript Chen et al., nicely describe the structure and functional characterization of a programmed -1 frameshift (-1 FS) and the intergenic (IGR) IRES in the Bemisia-associated dicistrivirus 2 (BaDV-2). The authors provide evidence that the predicted -1 FS actually promotes frameshift in vitro, using bicistronic reporter constructs. Similarly they also provide evidence for the initiation of translation driven by the IGR IRES but pointing to a molecular mechanism different from that described for other members of the dicistroviruses family. The authors combine structural and functional studies very well to show unambiguously the implication of these two “tools” of the viral genome in the regulation of the viral translation. However I have found several points that need to be considered.
Major points
- The experiments performed and, therefore the results are difficult to be understood without a clear and detailed diagram of the BaDV-2 dicistrovirus.
A schematic representation of the genome as provided in Figure 6B, but enlarged and including all the details necessary to follow the different experiments i.e., -1 FS with the hypothetical structure features; the IRG IRES, etc, should be provided as Figure 1, somewhere in the introduction.
RESPONSE: We thank the reviewer for this. We have now changed Figure 1 to provide more genome details. As for providing a more detailed structures for the IRES, we believe that given the order of experiments, 1st with the frameshifting then 2nd characterizing the IRES, showing the predicted FS structure first in Figure 1 helps guide the reader as we characterize the -1 FS signal first. Later, in the new Figure 4, we now provide the initial predicted IRES structure and the more elongated version (Figure 4A, 4B).
- The statistical analysis performed throughout the paper, the ANNOVA test, is not adequate for such a small sample size. ANNOVA is a parametric test, but for this sample size the authors should use a non-parametric test, for example, the Mann Withney test, or any other non-parametric test.
RESPONSE: Thank you for your invaluable feedback and for raising concerns about our statistical analysis, specifically the use of ANOVA given our sample size. We believe ANOVA remains a suitable analysis for our data.
Your suggestion to consider non-parametric tests for our sample size is well-received and was carefully considered. However, we would like to provide several reasons supporting our decision to use ANOVA:
Robustness of ANOVA to Assumptions: While ANOVA is a parametric test that typically benefits from larger sample sizes, it is also known for its robustness to homogeneity of variances, especially with equal sample sizes across groups. This characteristic is particularly relevant in our study, where we ensured equal numbers of biological replicates in each group.
Data Normalization and Transformation Procedures: We employed data normalization and transformation techniques to address potential deviations from normality prior to applying ANOVA analysis.
Selection of Biological Replicates: The biological replicates in our study were selected to ensure they are representative of their respective populations. Combined with our stringent experimental controls, this approach has minimized variability and enhanced the reliability of applying ANOVA in analyzing our results.
We understand the importance of rigorous statistical analysis and share your commitment to the highest standards of research integrity. Our decision to use ANOVA was made after careful consideration of these factors, ensuring that our findings are both robust and meaningful.
- Characterization of the BaDV-2 -1 FS signal.
It is difficult to follow the reasoning of the authors. First in the representation of the analyzed fragment they should indicate the coordinates of the real fragment they analyzed (3742-3884). Looking at the figure it seems that they are using a fragment of 3770-3865. The analysis of 3’ Del 1, 2 and 3 clearly indicate that just the deletion of the 21 nt at the 3’ end (3864 to 3865) results in a significant reduction of the -1 FS. But the authors do not give importance to this result. Instead they emphasize the inhibition by the Del 2 and Del 3 constructs. (page 5 line 219). The authors should discuss this issue. This result together with the very low frameshift efficiency of the full fragment, which has been quantified as only 4 and 8% depending on the extracts used, have the authors considered that nucleotides beyond 3884 might be necessary to achieve the 100% functionality of the -1 FS?
RESPONSE: We thank the reviewer for the comment and pointing this out. The WT -1 FS was from nucleotides 3742 to 3884, NOT as displayed in the original Figure 1. We apologize for this and have corrected this in the Figure 1. As for nucleotides beyond 3884, we are aware of long-distance interactions that may modulate -1 FS, however, with current understanding of -1 FS signals, our initial goal was to address the predicted -1 FS stem-loop/pseudoknot as this is the most likely FS region. As it is we have included 50 nucleotides downstream of the predicted -1 FS signal (3830-3884), to ensure we captured the FS signal. Based on our deletion analysis, we believe we have at least narrowed down the element that can support FS activity. Long distance interactions beyond 3884 will be examined in a future study.
- Figure 5A, the sequence cannot be read. Panel A of supplementary figure S1 should be included here as panel A. Then the current panel A should be panel B but replacing the current sequence representation of the analyzed fragment by a schematic line-drawing representation, clearly distinguishing the fragment corresponding to Fluc by a different line color or line shape. This scheme should include all structural and sequence details (e.g. by line colors necessary).
RESPONSE: We thank the reviewer for the comment. We believe you are referring to Figure 4A. We agree and have enlarged the fonts and provided a schematic line structure (no need to show nucleotides as in the original). Also, as suggested, we have included the predicted IRES structure from Figure S1 as Figure 4A. We now added the figure S1 schematic as figure 4A and updated the analyzed fragment to a schematic line-drawing representation. We believe this is much clearer. Thanks for the suggestion!
- Pg. 9 line 302. The authors state that 5’ Del1 and Del 2 inhibited IRES activity but figure 4 indicates no effect of 5’ Del1.
RESPONSE: We thank the reviewer for pointing this out. We now updated the manuscript (lines 338-339) to “Compared to the wild-type BaDV-2 IRES, 5’ deletions from nucleotide 5325-5421 (5’DEL 2) but not from nucleotide (nt) 5325-5366 (5’Del 1) inhibited IRES activity (Figure 4A), thus setting the boundaries from the 5’end of the BaDV-2 IRES.”
Line 309, combining 5’Del 2 and 3’Del 4 inhibited IRES activity to ~17% of wild-type IRES activity (Figure 4B). But in Figure 4B indicates ~40%.
RESPONSE: We thank the reviewer for the comment. We now updated the manuscript (lines 345-347) to “Combining 5’DEL 1 and 3’DEL 4 or 3’DEL 5 still showed IRES activity, whereas combining 5’DEL 2 and 3’DEL 4 inhibited IRES activity to ~40% of wild-type IRES activity (Figure 4B).”
- I agree with the authors that they have identified a 140 nt long fragment that supports translation initiation (IRES activity), but they cannot conclude that this is the minimal fragment or the minimal IRES. This statement should be carefully revised throughout the manuscript.
RESPONSE: We thank the reviewer for the comment. We agree and have changed all statements about the IRES to a “minimal” IRES (in quotes) and been careful to state that the “minimal” IRES is contained within the 140 nt element.
- Page 10 lines 341-344. This is confusing, are the authors using the same mutants included in Figure S1?, if so the nomenclature M1 to M7 does not correspond to that used in Figure S1.
RESPONSE: We thank the reviewer for the suggestion. The mutations made in figure 5 are new mutations for testing the SAXS-predicted structure but not from previous S1 mutations. We are now clarified the manuscript (lines 396-399) as “To examine whether parts of the secondary structure of SAXS IRES model is func-tions in translation, we re-examined the IRES translational activities by making muta-tions at the key predicted positions (Figure 5F, bottom).”
- It is difficult to accept that there is no significant differences for Mutants M1, M3, M7 and even M2 and M5. As mentioned in point 2, this analysis should be done with a non-parametric test.
RESPONSE: We thank the reviewer for the comment. Please see response to comment #2. For all data points, we have included all data from biological repeats of the experiments.
Minor points
- Introduction line 36. I understand that the authors mean: RNA viruses carry genomes with limited coding capacity; or RNA virus genomes have limited coding capacity.
Viruses, nor any other organisms encode a genome. Genomes encode information, not organisms encode genomes.
RESPONSE: We thank the reviewer for pointing this out. We now updated the manuscript to “ RNA viruses typically carry genomes with limited coding capacity.” (line 36)
- Page 2 line 69 delete subunit
RESPONSE: We thank the reviewer for pointing this out. We now deleted the extra “subunit” in line 69.
- Page 3 line 117, Nucleic Acids are not transfected. Cells are transfected with nucleic acids.
RESPONSE: We thank the reviewer for pointing this out. We now updated the manuscript to “S2 cells (3×106 cells) were transfected with bicistronic or infectious clone RNA (3 mg) using Lipofectamine 2000 reagent (Thermo Fisher Scientific).” (lines 121-122)
- Fig. 1B Nucleotide positions (numbers) should be indicated in the figure
RESPONSE: We thank the reviewer for pointing this out. We now added the nucleotide positions in the figure 1B.
- Figure 1C and 1D, lane numbers should be indicated in the gel.
RESPONSE: We thank the reviewer for pointing this out. We now added the lane numbers in figure 1C, 1D, and we also added lane numbers for figure 3A, 3B, 6C and 6D.
- Figure 2, Identification of panels B, C and D is missing.
RESPONSE: We thank the reviewer for pointing this out. We added the panels “B”, “C” and “D” to Figure 2.
- The meaning of n.s. should be indicated in all figures
RESPONSE: We thank the reviewer for pointing this out. We now added the meaning of “n.s” in all figure legends.
Figure 2A Constructs 3’ DEL are referred to in the text as Del. Inconsistency of the nomenclature
RESPONSE: We thank the reviewer for pointing this out. We now used the consistent format “DEL” when referring deletions in the manuscript.
- Figure legend 2D, line 260. The red dot line corresponds to the WT.
RESPONSE: We thank the reviewer for the comment. The red dot line corresponds to the WT is added but the one corresponds to the SS Mut also maintains as we want to show the background level.
- Page 10, line 341. “…are functional in translation”
RESPONSE: We thank the reviewer for pointing this out. We have altered to the sentence to “To examine whether the secondary structure predicted by the SAXS IRES model….” (line 396)
- Figure 5F nucleotide positions, coordinates, should be the same than in the previous figures. This new numbering makes very difficult to identify the mutations.
RESPONSE: We thank the reviewer for your comment. We now updated the nucleotide number in the figure 5F as consistent number as in the previous figures.
- Page 12 line 386 “evidence”
RESPONSE: We thank the reviewer for pointing this out. We now updated the manuscript to the correct spelling of “evidence”.
- KD should be Kd
RESPONSE: We thank the reviewer for pointing this out. We now changed all the “KD” to “Kd” in the manuscript.
- Figure 8, M2 has not been analyzed?
RESPONSE: We thank the reviewer for the comment. The M2 is analyzed when combined with the deletion of AUG (nt 5491-93).

Round 2
Reviewer 2 Report
Comments and Suggestions for Authors
No additional comments
Author Response
We thank the editor for the decision. Please find attached the revised manuscript with edits.
1) Most edits were grammatical and do not change meaning of sentences.
2) We added some sentences for clarification in figure legend and results such as the monocistronic description Lines 354-359 and in Figure Legend 3.
3) We replaced Figures 2, 3, 5 and 8 with cosmetics changes:
Figure 2: aligned the dotted red line in the graph.
Figure 3: added standard deviation bars for Figure 3C
Figure 5: aligned the dotted red line in the graph.
Figure 8: Enlarged the nucleotide numbers so it can be read easier.
Thanks!
